# Testing non-autonomous antimalarial gene drive effectors using self-eliminating drivers in the African mosquito vector *Anopheles gambiae*

David A. Ellis[ORCID][¤a¤b], George Avraam[ORCID], Astrid Hoermann[ORCID], Claudia A. S. Wyer[ORCID][¤c], Yi Xin Ong[ORCID], George K. Christophides[ORCID], Nikolai Windbichler[ORCID]*

Department of Life Sciences, Imperial College London, London, United Kingdom

¤a Current address: Ear Institute, University College London, London, United Kingdom
¤b Current address: Francis Crick Institute, London, United Kingdom
¤c Current address: Department of Life Sciences, Imperial College London, Silwood Park, Ascot, United Kingdom
* nikolai.windbichler@imperial.ac.uk

**Data Availability Statement:** The amplicon sequencing data from this study are available in the

## Abstract

Gene drives for mosquito population modification are novel tools for malaria control. Strategies to safely test antimalarial effectors in the field are required. Here, we modified the *Anopheles gambiae zpg* locus to host a CRISPR/Cas9 integral gene drive allele ($zpg^D$) and characterized its behaviour and resistance profile. We found that $zpg^D$ dominantly sterilizes females but can induce efficient drive at other loci when it itself encounters resistance. We combined $zpg^D$ with multiple previously characterized non-autonomous payload drives and found that, as $zpg^D$ self-eliminates, it leads to conversion of mosquito cage populations at these loci. Our results demonstrate how self-eliminating drivers could allow safe testing of non-autonomous effector-traits by local population modification. They also suggest that after engendering resistance, gene drives intended for population suppression could nevertheless serve to propagate subsequently released non-autonomous payload genes, allowing modification of vector populations initially targeted for suppression.

## Author summary

Gene drive is a method that allows the genetic modification of entire populations of harmful organisms. Their application to tackle invasive species, agricultural pests or insect disease vectors has been suggested. For example, they could reduce the capacity of malaria mosquitoes to transmit this deadly disease to humans by producing effector molecules inhibiting the development of the *Plasmodium* parasite in the mosquito vector. We describe a strategy to modularize and test multiple transgenes destined for release, and to introduce only the minimal set of modifications needed into a mosquito population. We show how some elements, once no longer needed, can be made to self-eliminate from populations and we also study how several independent gene drive traits, located in

European Nucleotide Archive under the study
accession PRJEB50687.

**Funding:** This work was supported by a Bill and
Melinda Gates Foundation grant (OPP1158151) to
GKC and NW. The funders had no role in study
design, data collection and analysis, decision to
publish, or preparation of the manuscript.

**Competing interests:** The authors have declared
that no competing interests exist.

different parts of the genome, can interact and propagate at the level of mosquito cage populations.

## Introduction

The encouraging gains to curb malaria made since the beginning of the millennium have levelled off in recent years, and novel tools for the eradication of malaria in high-endemic areas are urgently needed [1]. Gene drive systems based on the genetic modification of vector populations are a promising set of tools conceived to either eliminate mosquito populations or reduce their ability to transmit the *Plasmodium* parasite to humans. Homing-based drives, first proposed in the 2000s [2] and now based on CRISPR [3,4], have demonstrable potential to reduce the size of mosquito populations (population suppression) or to render them ineffective as vectors of disease (population replacement or modification).

Population modification requires the expression of molecules that can efficiently disrupt *Plasmodium* development within the mosquito vector. A number of endogenous and exogenous effector molecules have been expressed in transgenic mosquitoes to interfere with parasite development, including the antimicrobial peptide (AMP) Cecropin A [5], fibrinogen domain-containing immunolectin 9 (FBN9) [6], thioester-containing protein 1 (TEP1) [7], the NF-kappaB transcription factor Rel2 [8], as well as the protein kinase Akt [9,10] and the Phosphatase and Tensin homolog (PTEN) [11]. Mosquitoes expressing exogenous synthetic peptides Vida3 [12] and SM1, engineered as a quadruplet [13], the bee venom phospholipase A2 (PLA2) [14,15], single-chain antibodies (scFvs, m1C3, m4B7, m2A10) targeting parasite proteins [16,17], and more recently, microRNA sponges [18] have also shown promise at reducing *Plasmodium* transmission. Progress has also been made optimising gene drive systems themselves, with a number of key conceptual or experimental advances reducing the formation of target site resistance [19–22] and limiting drives, spatiotemporally [23–27].

Although gene drives hold great promise, their ability to alter wild populations could have undesirable consequences of an ecological, economic, or even medical nature. Because of this, approaches are required to safely test such systems, particularly novel antimalarial effector molecules, at an ever-larger scale, moving in defined steps from the lab to the field. Lab containment has been discussed previously [28], and some projects are now considering field containment, i.e., geographical isolation [29,30]. Regarding the design of gene drive systems, different self-limiting approaches have been proposed (e.g. [31]). Modular or split gene drives–where a non-autonomous gRNA-only effector transgene cannot drive in the absence of an unlinked Cas9-expressing transgene–have emerged as a promising approach to mitigate risk [23–26,32–40]. In such systems, the gradual loss of a non-driving Cas9 transgene would inherently limit the spread of dependent effector transgenes over time. However, such Cas9 elements may persist in populations, albeit at a low frequency. An additional safeguarding step could use a Cas9 source that is actively removed from the population. When gene drives were conceived, it was suggested that costly drives could be recalled by introducing resistant alleles to a population, which would then be strongly selected at the expense of the drive [2]. The impact of resistant alleles on gene drive has since been shown experimentally [19,41] and widely discussed [42]. However, this does not apply to non-driving Cas9 transgenes which cannot trigger their own replacement by resistant alleles (although note that if costly, they should still disappear by natural selection). An autonomously driving, costly Cas9-source, with pre-existing resistant alleles, could present an alternative approach to actively remove the Cas9 source from a population. Such a system would leave a smaller ecological scar, spreading only

those genetic modifications necessary for control through a population. It may also reduce or sidestep strict requirements for geographical isolation.

We have developed a gene drive framework that allows safeguarding and possibly reduces the time between lab and field testing [37]. Termed integral gene drive (IGD), this approach integrates effector molecules within endogenous mosquito loci, hijacking their gene-expression profiles. In anticipation of regulatory hurdles, IGD reduces the complexity and size of genetic modifications required for each transgene. After safe testing of the engineered transmission blocking traits in the absence of gene drive, these transgenes can be mobilized into non-autonomous gene drives when a source of Cas9 is provided in *trans*. We have recently demonstrated the efficacy of a handful of such non-autonomous transgenes [43].

The benefits of multiplex genome engineering for rapidly making complex or combinatorial genetic modifications has been widely reported [44–46]. Given the vast array of antimalarial molecules and genetic loci at which to express them, the multiplexing of effectors is an attractive approach for population modification strategies [47]. Multiplexing could have considerable benefits, from increasing throughput during initial testing (where population trials are a considerable bottleneck), to expanding the repertoire of antimalarial effectors spreading through a population in an approach analogous to combination therapy [48]. However, whether multiple effector transgenes can be safely driven through mosquito populations simultaneously has not been tested.

As a set of steps for moving safely from the lab to the field, we previously proposed the use of a non-driving Cas9 transgene as a first step, allowing a limited form of gene drive (self-limiting), before moving to an autonomously driving Cas9 transgene and full gene drive (self-sustained) [37]. Here, we sought to design and test an IGD at the *zpg* locus capable of providing the antimalarial transgenes with a source of Cas9. We show that an autonomously driving Cas9 at *zpg*, with large costs in a susceptible genetic background, can also be effectively used as a self-limiting drive when resistant alleles are present. By combining this transgene, named *zpg^D*, with non-autonomous effectors and multiplexing the individuals in four distinct caged populations, we were able to study the dynamics and mutation profiles of multiple effector transgenes simultaneously. We show that *zpg^D* can effectively drive multiple payloads through a population but is selectively removed over time. This approach has the potential to leave mosquito populations with minimal genetic modifications, limiting changes to those necessary for improving global health, through a combined minimal design of the effector transgenes and active removal of Cas9.

## Results

### Direct integration of gene drive modules at the *zpg* locus

The promoter region of the *zero population growth* gene (*zpg*) has been used successfully in conventional gene drives, providing effective Cas9 expression in *A. gambiae* germline [21,22]. Therefore, we chose *zpg* as a target locus for the generation of an IGD [37,43]. We designed gene drive constructs for integration at the start of the *zpg* coding sequence (CDS; **Fig 1A**). These constructs contained a gRNA within an artificial intron introduced into the Cas9 CDS and a 3xP3::mScarlet fluorescent marker cassette, located on the antisense strand and designed for sense-strand read-through translation linked via a 2A signal to both the Cas9 and the endogenous zpg CDS. Such an approach has not been previously tested and is an alternative to marker excision, a strategy we have previously used to minimize the effects of modifications on the targeted host gene. Upon integration, the transgene would therefore co-opt the *zpg* regulatory regions and express Cas9 in the male and female germline, which would result in two possible outcomes: read-though translation that generates sufficient *zpg* gene product, while

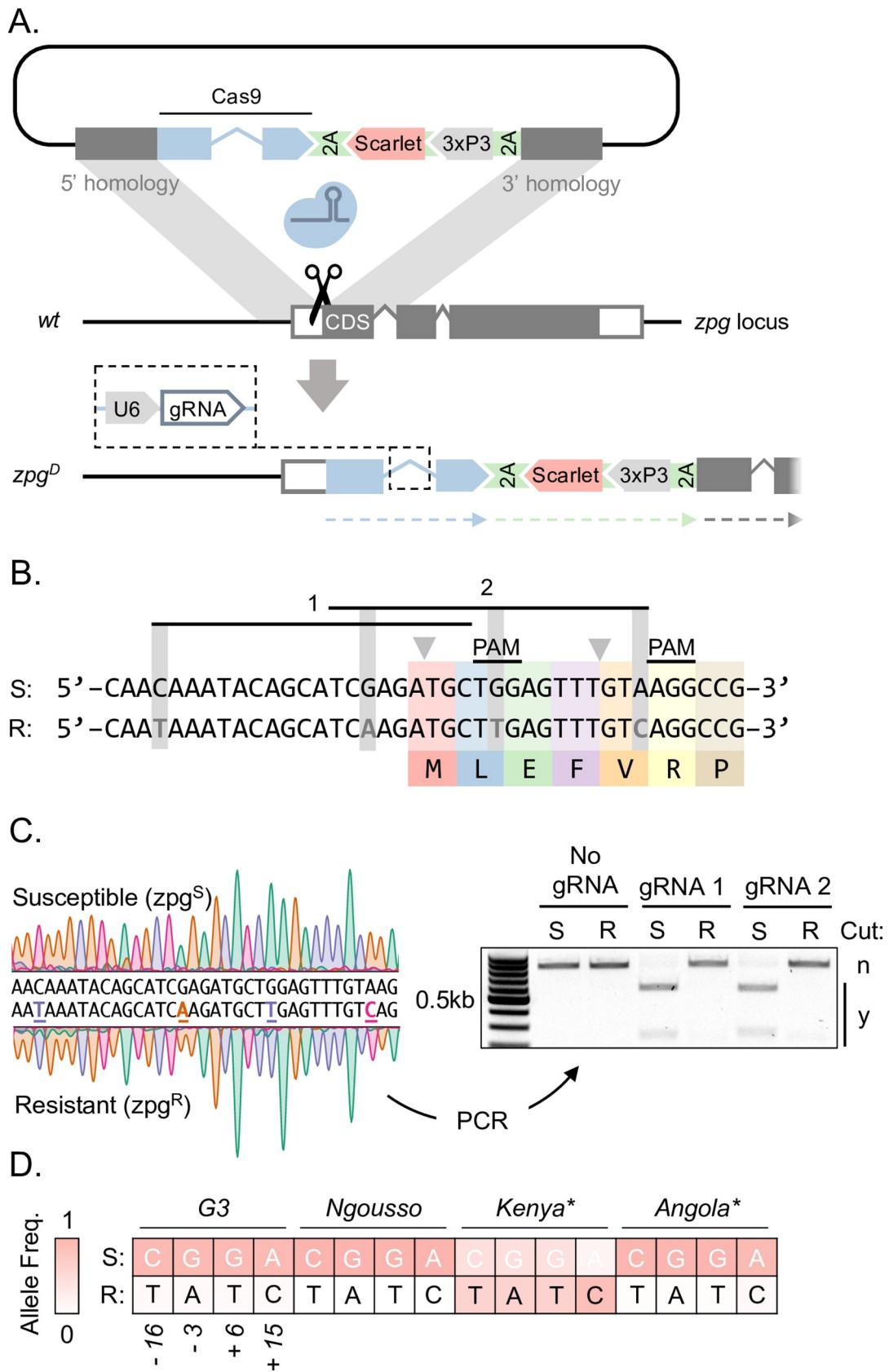

**Fig 1. Gene drive integration at the *zpg* target locus and pre-existing target site variants. A**: Transformation construct with the Cas9 integral drive cassette designed to target the *zpg* locus near the start codon for N-terminal integration. The resulting *zpg^D* allele and the intronic gRNA are shown on the bottom. The predicted separate gene products following translation and ribosomal skipping (dashed arrows beneath), including Cas9 and zpg are indicated. **B**: Sequence of the segregating susceptible (*zpg^S*) and resistant (*zpg^R*) haplotypes with respect to the two gRNA designs used. The first 7 *zpg* codons are indicated. PAM; protospacer adjacent motif. Nucleotides highlighted in grey are SNPs. **C:** Molecular analysis of the two naturally occurring *zpg* alleles present in the G3 lab colony. Sanger sequencing traces of PCR products from each background and *in-vitro* cleavage of PCR products by gRNA1 and gRNA2. PCR products from *zpg^S* and *zpg^R* abbreviated to S and R, respectively. **D:** Allele frequencies of the 4 SNPs constituting the R allele in both wild (asterisk) and lab populations of mosquitoes. Allele frequencies were estimated from Illumina whole genome sequencing of 24 G3 individuals, Sanger sequencing at the *zpg* locus of 24 Ngousso individuals, as well as Illumina WGS from the Ag1000G project.

allowing the construct to behave as a neutral IGD or, in case of a significant reduction or loss of *zpg* host gene function, a construct that impairs or disrupts mosquito fertility, since *zpg* function is important for gonad development in both males and females [49]. In the latter case, if homing occurred in the germline, it could cause a dominant loss of *zpg* expression already in hemizygous individuals.

An analysis of targetable sites near the *zpg* start codon yielded 2 possible gRNA designs for Cas9 (**Fig 1B**) with partially overlapping (9bp) target sites and cut sites located either within (gRNA 1) or 9 base pairs downstream of the start codon (gRNA 2). An analysis of genetic variation at the target sites revealed an alternative *zpg* allele with a number of single nucleotide polymorphisms that would be expected to impair Cas9 activity. Using *in-vitro* cleavage experiments, we confirmed the activity of both gRNAs against the wild-type susceptible target allele (*zpg^S*) and that the presence of the resistant allele (*zpg^R*) abolishes Cas9 activity *in-vitro* (**Fig 1C**). An analysis of population genetic data of lab strains in conjunction with wild-caught mosquitoes of the AG1000 dataset [50] suggested that the alternative R haplotype is present at varying frequencies in these populations (**Fig 1D**). For example, the R allele is absent in Angola but predominates in Kenya. In the G3 lab strain we detected the R allele at low levels (<10%). For experimental evaluation we established 2 transgenic strains utilising the cassette described above, one for each gRNA (*zpg^D_{gRNA1}* and *zpg^D_{gRNA2}*), by CRISPR-mediated knock in using a plasmid with ~2kb homology arms flanking the cassette (**Fig 1A**).

## Transmission of *zpg^D* and a payload in susceptible and resistant genetic backgrounds

We first crossed, in individual mating pairs, either male or female *zpg^D* hemizygotes carrying gRNA1 or gRNA2 with wild-type G3 mosquitoes and scored the frequency of transgenic progeny via the presence of the mScarlet (RFP) marker. To determine their zpg^R or zpg^S status, hemizygous parents were then genotyped by genomic PCR and Sanger sequencing. For both constructs, we found Mendelian inheritance of the *zpg^D* allele when it was paired with the *zpg^R* allele irrespective of the sex of the transgenic parent (**Fig 2A**, purple bars). This corroborates the *in-vitro* experiments and suggests that the *zpg^R* allele is refractory to both gRNAs *in vivo*. When, in transgenic males, the *zpg^D* allele was paired with the susceptible *zpg^S* allele, we observed super-Mendelian inheritance of both constructs with transmission rates of 96% for gRNA1 and 76% for gRNA2, respectively (**Fig 2A**, orange bars). In contrast, no progeny were obtained from the reciprocal crosses, i.e., of transgenic *zpg^D/zpg^S* females carrying either gRNA to wild-type males (**Fig 2A**). We concluded from this set of results that *zpg^D* alleles produce functional Cas9 and gRNA and are able to initiate homing efficiently if a targetable site is present on the homologous chromosome. We also concluded that homing was able to induce homozygosity of *zpg^D* in the germline and leads to a loss of fertility in females, but not in

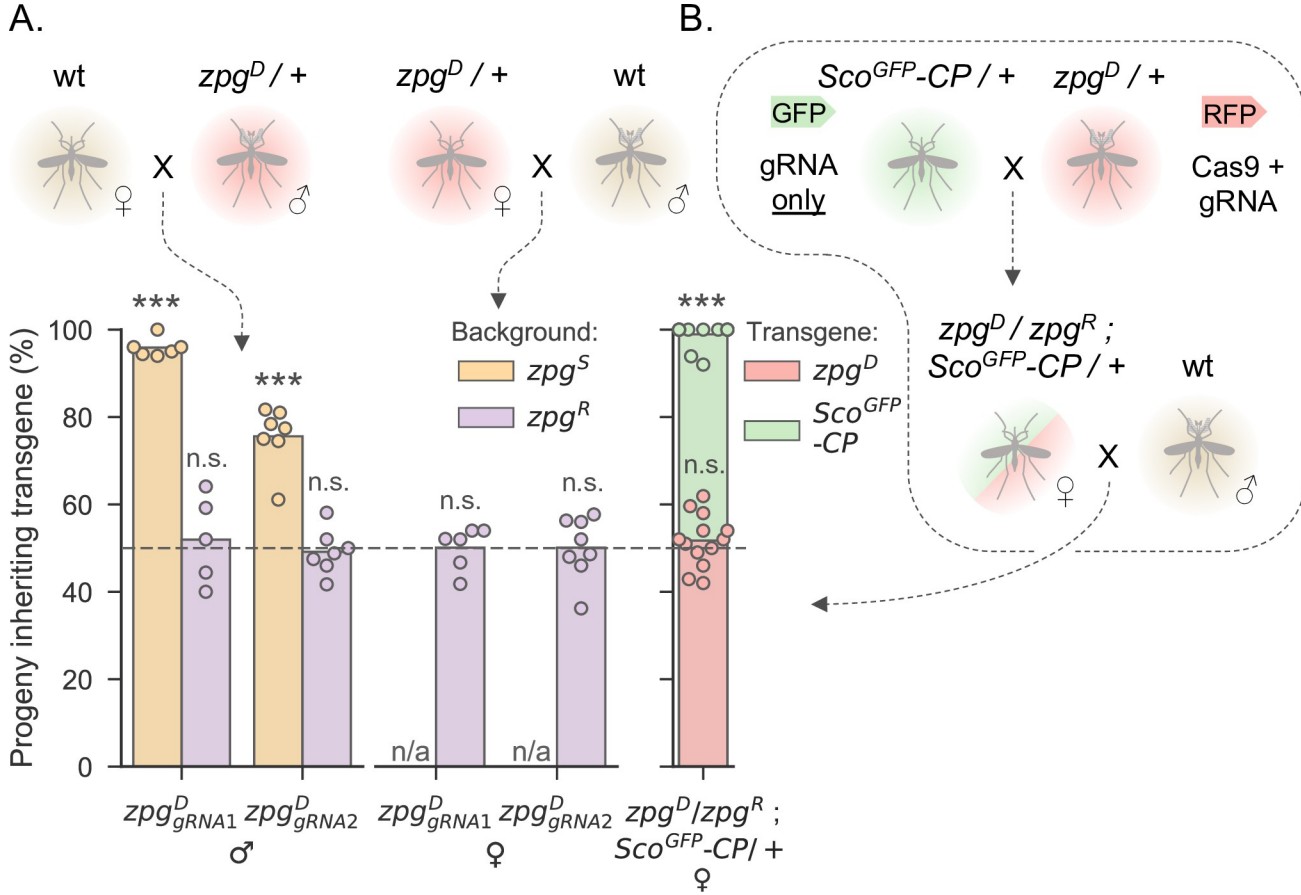

**Fig 2. *zpg^D* produces functional Cas9 and homes in a susceptible genetic background, but *zpg^D*/*zpg^S* females do not produce offspring. A**: Proportion of progeny inheriting the *zpg^D* transgene from single crosses between hemizygous and wild-type mosquitoes. *Left*: Hemizygous males expressing gRNA1 or gRNA 2 crossed to wild-type females. *Right*: Hemizygous females expressing gRNA1 or gRNA 2 crossed to wild-type males. The transgenic parent from each cross was genotyped by Sanger sequencing to determine the genetic background of the transgene in that individual–those with susceptible alleles (*zpg^S*) are shown in orange, whereas those with resistant allele are in purple (*zpg^R*). Males expressing both gRNA1 and gRNA2 in a susceptible background sired significantly more transgenic progeny than 50% (gRNA1: t = 51.861, p<0.0001; gRNA2: t = 9.749, p<0.0001; one sample t-tests). In crosses with hemizygous females, none of the females producing viable progeny had a *zpg^S* genetic background. **B**: Female *zpg^D* individuals produce active Cas9 that allows non-autonomous homing of un-linked construct in a *zpg^R* background. The progeny of a cross between wild-type males and females hemizygous for *zpg^D* and a non-autonomously homing transgene at the *cp* locus (Sco^GFP^-CP) were screened for the inheritance of each transgene. Females sired significantly more Sco^GFP^-CP progeny than 50% (t = 65.913, p<0.0001; one sample t-test).

males. We employed *zpg^D_gRNA1* for all subsequent experiments, hereafter simply referred to as *zpg^D*.

To corroborate the conclusion that hemizygote *zpg^D*/*zpg^S* female infertility is the result of homing in the germline, and to understand how *zpg^D* would interact with other gene drive elements, we generated transhemizygous females carrying *zpg^D* in combination with a *zpg^R* allele, as well as the previously characterized non-autonomous payload drive Sco^GFP^-CP, which harbours a gRNA targeting the CP wild-type allele but lacks Cas9 [43]. This non-autonomous drive element expresses the antimicrobial peptide Scorpine [51] from within the CP host gene which is strongly induced upon bloodfeeding. When crossed to wild-type males in single mating pairs we found that, as observed previously, *zpg^D* was transmitted in Mendelian fashion to the progeny. In contrast, Sco^GFP^-CP was transmitted to nearly all progeny (99%; **Fig 2B**). This shows that females carrying a *zpg^D* allele are able to produce functional Cas9, further supporting our conclusions that *zpg^D*/*zpg^S* female infertility is a result of homing in the female

germline, and that the inability of $zpg^D$ to home in a $zpg^R$ background is not because of a lack of functional Cas9 expression. It also shows that $zpg^D$ can provide Cas9 in trans for efficient non-autonomous homing at other loci.

## Fertility and gonad development of $zpg^D$ transgenics

To better understand the effects of $zpg^D$ integration at the $zpg$ locus, we crossed hemizygous males to hemizygous females, obtaining a pool of progeny with different allelic combinations. Since only $zpg^D$/$zpg^R$ hemizygous females are fertile (**Fig 2A**), no progeny in this pool came from $zpg^D$/$zpg^S$ mothers. First, using primers annealing to genomic regions external to the homology arms of the template plasmid, we performed long-range PCR to genotype individuals from this pool (**Fig 3A**, primer set 1 and gel image). In this experiment, both hemizygous and homozygous individuals were identified in both sexes, showing that all genotypes are viable.

   To better understand the effects of different allelic combinations at the $zpg$ locus on male and female fertility we measured life history and physiological traits in a set of individuals from this pool. In single pairs, we crossed individuals to the wild-type and recorded: 1) whether the copulation was successful (based on whether or not the female laid viable eggs, and if they didn't, the status of their dissected spermatheca), and 2) the number of offspring that were sired. 46 male crosses and 46 female crosses were analysed in total. We used PCRs to genotype individuals, and sequenced $zpg$ alleles of those that were hemizygous (**Fig 3A**, primer sets 2 and 3; see methods). The following were successfully genotyped: hemizygous females n = 24; hemizygous males n = 33; homozygous females n = 17; homozygous males n = 10. Of the 33 crosses with $zpg^D$/$zpg^R$ fathers, 12 led to successful insemination of females (36%), whereas of the 10 crosses with $zpg^D$/$zpg^D$ fathers, only 2 were successful (20%). RNA interference (RNAi)-mediated knock-down of $zpg$ has been previously reported to lead to completely sperm-less males [52]. Whilst we found no significant difference between the insemination rate of hemizygous and homozygous $zpg^D$ males ($X^2 = 0.936$, p>0.05), the low insemination rates, particularly of $zpg^D$/$zpg^D$ males, suggest $zpg^D$ may display some phenotypic manifestation of reduced zpg expression. Next, we analysed the number of offspring sired by each genotype. Whilst hemizygous ($zpg^D$/$zpg^R$) females produced an average of 79 ± 26 offspring (mean ± standard deviation)–a similar number to hemizygous males (74 ± 34)—from those homozygous ($zpg^D$/$zpg^D$) females that mated and laid eggs, none generated offspring (**Fig 3B**). This was not the case in homozygous males where viable progeny were produced (26 ± 18, mean ± standard deviation). Due to their unequal sample sizes, a Welch's independent two-sample t-test was used to compare the number of offspring sired by the two male genotypes (Welch's t(df = 2) = 3.035, p = 0.070), although non-significance is probably due to the low number of homozygous males. Whilst there may be some impact on male fertility, viable progeny were still sired by both genotypes in males, suggesting that $zpg^D$ maintains some expression of functional $zpg$ gene product in males. We examined gonad morphology of male and non-blood-fed female mosquitoes with different genotype combinations. Pronounced atrophy of the ovaries was observed in both $zpg^D$/$zpg^S$ and $zpg^D$/$zpg^D$ females (Fig 3C). This was also quantified using measurements along the major axis (anterior-posterior) of each ovary (**Fig 3D**; t(df = 24) = 8.342, p<0.0001, and t(df = 22) = 7.988, p<0.0001, respectively; independent two-sample t-tests comparing each genotype to wild-type). In homozygous $zpg^D$ males, testes physiology and contents appeared slightly abnormal with the area of testis marginally smaller than wild-type males (t(df = 3) = 3.053, p = 0.055; independent two-sample t-test). The other genotypes however, appeared to have testes of normal morphology, and all genotypes appeared to have testes of similar lengths (**Fig 3C and 3D**; df = 2 in all cases). Fig 3E summarizes our observations regarding the effects on fertility of various allele combinations at the $zpg$ locus.

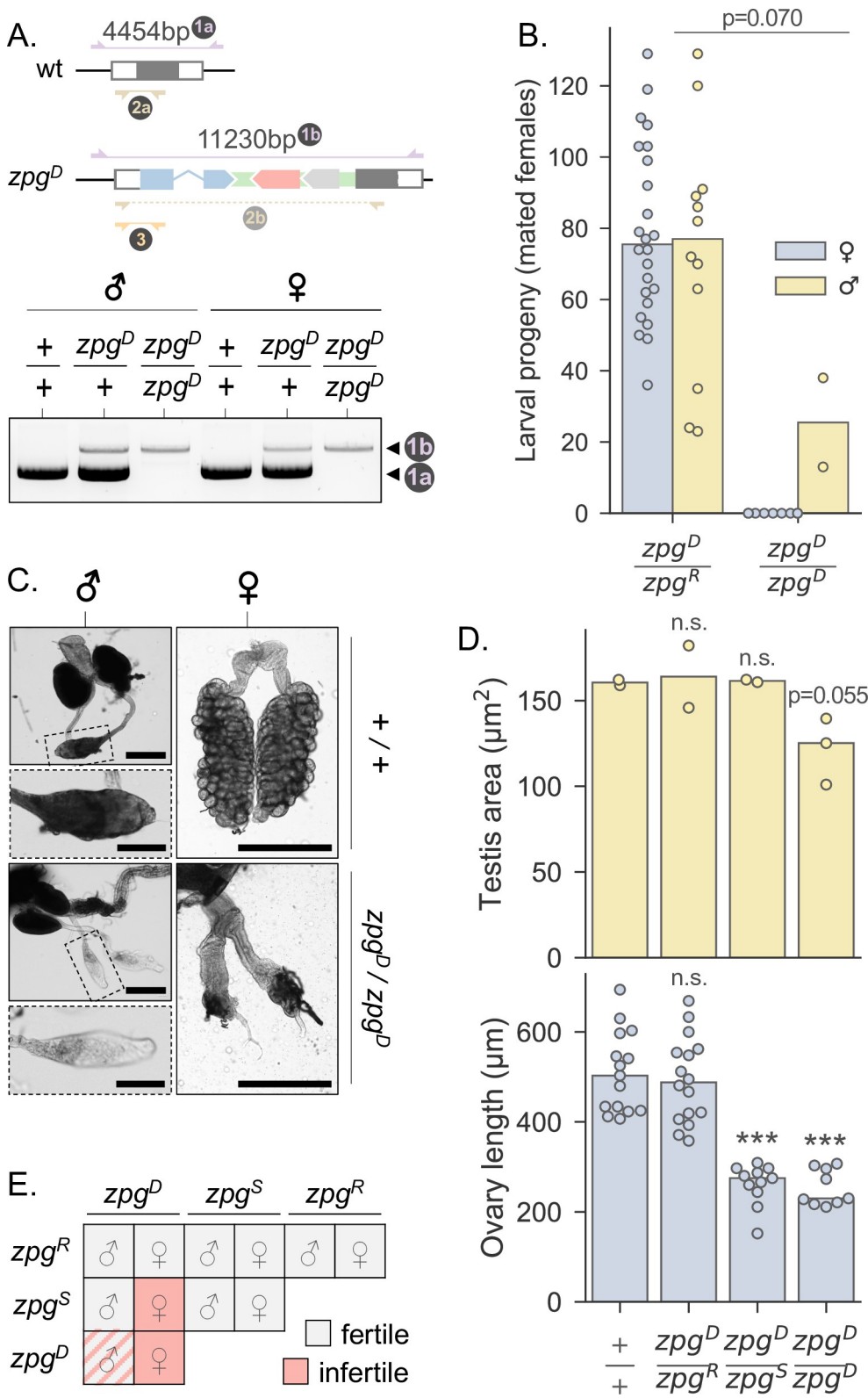

**Fig 3. _zpg^D_ homozygotes are viable, but germline homozygosity results in sterility. A**: Molecular PCR assay to differentiate the _zpg_ genotypes of adult mosquitoes. Primers in set 1 (purple) bind outside the original homology arms. These were used to confirm that _zpg^D_ had integrated at the endogenous gene, and to identify adult wt, heterozygous

and homozygous individuals as indicated in the exemplary gel below. Primer sets 2 (brown) and 3 (yellow) were additionally used for large-scale screening. **B**: The offspring of a cross between hemizygous $zpg^D$ males and females were themselves crossed to wild-type males or females in cups, before being genotyped. The number of larval progeny from these crosses for which mating was confirmed is shown. 46 males and 46 females were analysed in total. Of those, the following were successfully genotyped: hemizygous females n = 24; hemizygous males n = 33; homozygous females n = 17; homozygous males n = 10. **C**: Example images of dissected wild-type and homozygote male testes (left) and non-blood-fed female ovaries (right). For males, an overall image of the entire reproductive tract is shown (scale bar 200μm) with a cutaway of a single testis (scale bar 100μm). For females, entire ovaries and reproductive tracts are shown (scale bars 400μm). **D**: Male and female mosquitoes of the indicated $zpg$ genotypes were dissected and testis area and ovary length determined. Male testis area is shown in yellow and female ovary length in blue. Comparing ovary lengths to wild-type (n = 15) using independent two-sample t-tests, both homozygote (t = 7.988, p<0.0001, n = 9) and $zpg^D$/$zpg^S$ (t = 8.342, p<0.0001, n = 11) females differed significantly, whereas $zpg^D$/$zpg^R$ females did not (t = 0.515, p = 0.610, n = 16). Comparisons of male testis area to wild-type (n = 2) showed that only homozygotes approached significance (t = 3.053, p = 0.055, n = 3), and not $zpg^D$/$zpg^S$ (t = -0.537, p = 0.645, n = 2) or $zpg^D$/$zpg^R$ (t = 0.242, p = 0.831, n = 2). **E**: Summary of the observed fertility effects of $zpg$ allele combinations.

## Population cage dynamics of $zpg^D$ in combination with payload drives

We observed that, in females, the non-autonomous drive element Sco$^{GFP}$-CP could be transmitted to progeny at high rates even when $zpg^D$, which provides Cas9, encountered target site resistance at the $zpg$ locus (**Fig 2B**). This outcome suggests that the presence of resistance alleles would have different effects on $zpg^D$ and on any non-autonomous gene drives it could mobilize i.e. resistance alleles would aid the rapid replacement of the Cas9 driver locus, whilst also allowing more gene drive to occur at the effector loci. Self-eliminating drivers could thus enable local population modification without leaving behind any driver element. We examined this effect in multiplexed population cage experiments of *A. gambiae* where transhemizygous individuals carrying diverse effectors were released simultaneously in each cage. To model the expected dynamics of each transgene over consecutive generations, we employed an agent-based model (**S1 Methods**) where we considered populations of 600 individuals with discrete generations. We investigated the interaction of $zpg^D$ (starting allele frequency 15%) and 3 previously characterized non-autonomous and markerless payload drives [43]: Sco-CP, *Aper1-Sco* and ScoG-AP2 (starting allele frequencies of 5%, respectively). The model considers the observed fertility effects and allows for the inclusion of pre-existing target site resistance, as well as random *de-novo* generation of target site resistance at each of the 4 loci. It also simulates random sampling from each cage population with nonoverlapping generations.

First, we analysed a scenario where populations were seeded with the allele frequencies described above, but every alternative allele at the $zpg$ locus was susceptible ($zpg^S$). We found that, despite active gene drive in males, no significant net change to the frequency of the payload drives occurs over time with $zpg^D$ declining in frequency (**S1 Fig, panels A,B**). This most likely reflects the transgenes being passed on by nearly all males, but nearly no females, who when hemizygous for an effector are also most likely $zpg^D$/$zpg^S$ and therefore sterile. Next, we modelled the impact of introducing pre-existing target site resistance (5% starting allele frequency of $zpg^R$). While rapid self-elimination of $zpg^D$ was still predicted, we found that the introduction of resistant alleles would induce gene drive and increase effector frequency at the 3 payload loci (**S1 Fig, panels C,D**). This was due to active drive of the payloads now also occurring in zpg$^D$/zpg$^R$ females, which unlike zpg$^D$/zpg$^S$ females, were completely fertile.

We sought to experimentally validate these predictions in 4 independent cage experiments. To facilitate our experiments, we established two G3 wild-type populations which contained only the $zpg^S$ or the $zpg^R$ allele. Each cage was initiated using 120 and 300 $zpg^S$/$zpg^S$ males and females, respectively; 60 $zpg^D$/$zpg^S$, Sco-CP/+ males; 60 $zpg^D$/$zpg^S$, ScoG-AP2/+ males; as well as 60 $zpg^D$/$zpg^R$, *Aper1-Sco*/+ males (see Methods). Each generation we sampled and genotyped 47 random individuals by multiplex PCR. This was done as the effector constructs used

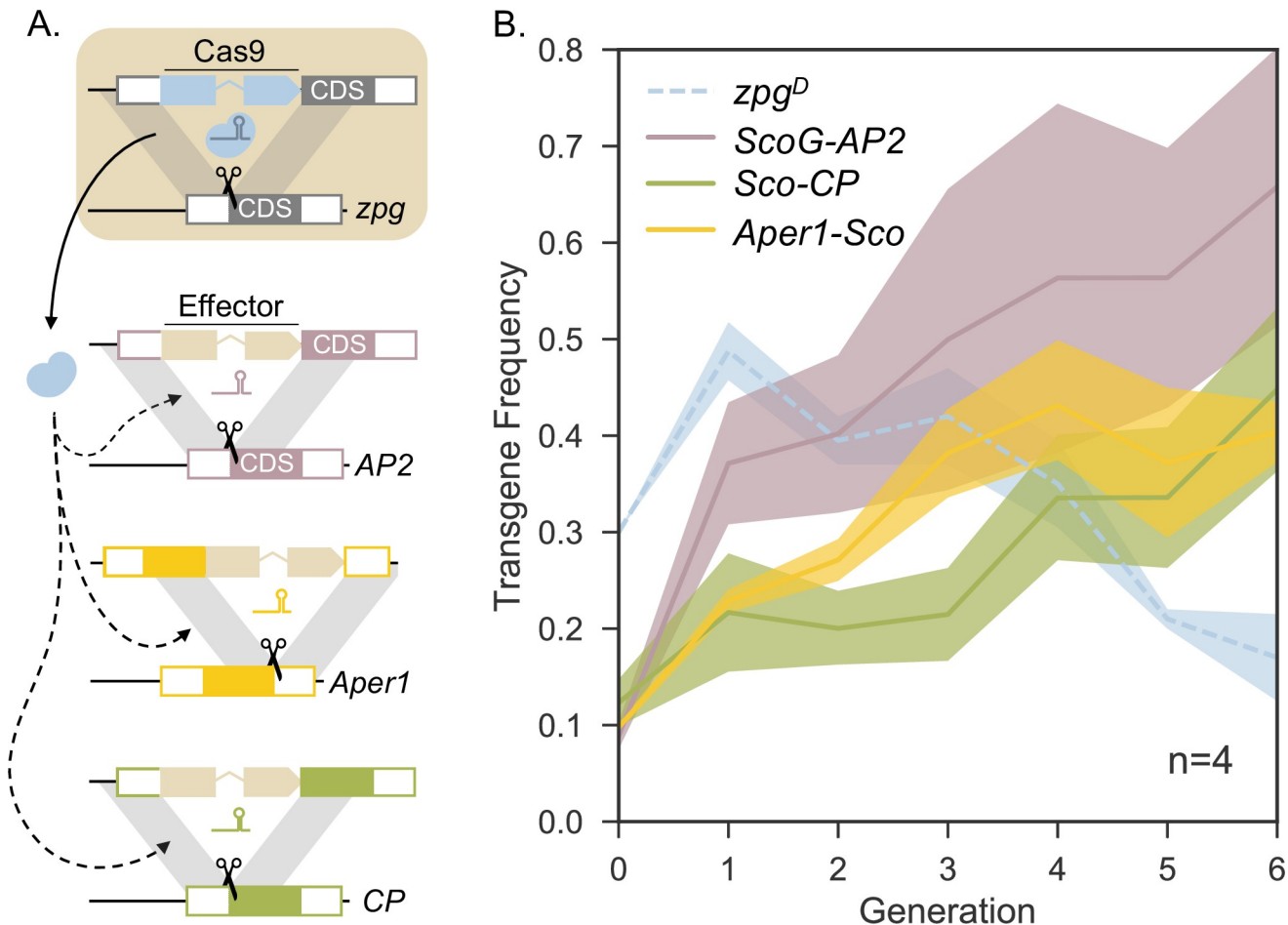

**Fig 4. *zpg$^D$* self-eliminates but drives the spread of three non-autonomous payload genes across four independent caged populations. A:** Illustration of *zpg$^D$*-dependent homing at the three non-autonomous integral gene drive loci. The germline-expressed Cas9 forms a complex with U6-driven gRNAs expressed by and specific to each effector construct. **B:** Transgene frequency of each construct estimated by either fluorescence (*zpg$^D$*) or multiplex PCR. Solid lines and shaded areas indicate the means ± 68% bootstrap CI respectively for zpgD (blue), ScoG-AP2 (purple), Sco-CP (green) and *Aper1-Sco* (yellow). 47 individuals and one negative control were genotyped per cage and per generation. Cages contained ~600 individuals in total.

in these cage trials did not carry phenotypic markers [43] and because visible markers rapidly lose their utility when many constructs are in play. We observed a decline in the egg output in all 4 cage populations, peaking at <45% in generation 2 (**S2 Fig**). With the decline of *zpg$^D$* to an average transgene frequency of <20% in generation 6, egg production recovered in all populations. Across the 4 cage populations the transgene frequency of the 3 non-autonomous effector traits increased over the course of the experiments, in line with the predictions of the model (**Fig 4**).

## Analysis of target site alleles in the caged populations

Following cleavage by Cas9 in the germline, DNA double-strand breaks (DSB) can be resolved by the DNA repair machinery in several ways. Homology directed repair (HDR) using the drive allele as a repair template leads to conversion of a wild-type allele to a drive allele (i.e. homing). Due to the design of gene drives, this outcome eliminates the gRNA target site making the new drive allele resistant to future cutting. However, DSB can also be repaired via DNA end-joining pathways such as non-homologous end joining (NHEJ) or microhomology-

mediated end joining (MMEJ) which does not lead to drive conversion. The resulting small deletions or insertions generally destroy the gRNA target site and abolish recognition by Cas9. Such cleavage resistant alleles, which can also occur naturally as standing genetic variation of a population (as is the case for zpg$^R$), present a barrier to gene drive. These mutations may also arise outside of the germline in developing embryos as a result of maternally deposited Cas9 [19,20,22,41]. With respect to target gene function, the impact of such mutations can range from being detrimental (termed R2 alleles) to being neutral, for example when they result in synonymous substitutions or small tolerable deletions that maintain the reading frame (termed R1 alleles). R1 alleles that present a lower relative cost than a circulating drive will therefore undergo strong selection [19,41,42,53,54]. To better understand the observed cage dynamics and assess the level to which resistance may evolve at each locus, we performed amplicon sequencing of the target sites of all 4 host genes at generations 1 and 6 in each of the 4 replicate cage populations. This was performed on a pool of 100 mosquitoes for each sample and PCR conditions were chosen to amplify wild-type-sized alleles potentially modified by NHEJ or MMEJ but not the respective transgenes at each locus. We observed a significant increase in alleles predicted to impair or abolish gRNA recognition at 3 out of the 4 loci (**Fig 5A**). At the *zpg* locus >60% of all reads in generation 6 represented resistant alleles–a 1.3-fold increase from generation 1. The pre-existing R1 resistant allele we had characterized and introduced was the most predominant, increasing 3-fold from 7% at G1 to 21% of all reads at G6 (**Fig 5B**). At both the *aper1* and *ap2* loci, resistant alleles made up 30% and 19% of all reads at generation 6 (2.9- and 1.7-fold increases), respectively. In both cases single, predominant R1 alleles also present at generation 1 and carrying mutations relatively distant from the predicted cleavage site made up the majority of the resistant alleles (Fig 5B). Interestingly, there was no significant increase in resistant alleles at the *cp* locus. The top 5 most frequent resistant alleles, all predicted to maintain CP expression, accounted for <1% of reads on average and overall, less than 10% of reads differed from the wild-type *cp* target sequence in either generation 1 or 6.

## Discussion

By integrating a small number of genetic features at endogenous genomic loci and hijacking their gene-expression machinery, the amount of genetic information that needs to be introduced to a transgenic organism can be greatly reduced. Along with the excision of fluorescent markers, this could create minimal and modular gene drive elements for population modification [37,43]. Here, we integrated the CDS of Cas9, including a synthetic gRNA-containing intron, at the *zpg* locus with the aim to establish an autonomous integral gene drive. Our results suggest that this allowed for high rates of homing of the modified allele (termed *zpg$^D$*) in males and females and of non-autonomous transgenes at separate loci each expressing a respective cognate gRNA. However, homozygosity of *zpg$^D$*, and probably homing-induced homozygosity in the germline of hemizygous females, led to fertility defects. Note that some low level somatic "leakiness" has been reported when using the *zpg* promoter for germ-line expression [22], and hemizygote infertility could also result from expression of *zpg$^D$* in the soma. Both *zpg$^D$*/*zpg$^D$* and *zpg$^D$*/*zpg$^S$* females were sterile and lacked fully developed ovaries whereas homozygous *zpg$^D$*/*zpg$^D$* males showed low levels of fertility. Whilst the reasons for this are unclear, 2A skipper peptides have been shown to reduce the expression of subsequent CDS due to inefficient cleavage [55,56], and the *zpg$^D$* Cas9 module was linked with the endogenous *zpg* CDS by a 2A-separated translational unit representing the fluorescent transformation marker. This was designed for read-though translation, where as well as Cas9, the ribosome would translate the reverse complement and coding sequences of the marker, followed by the *zpg* host gene product. However, our results suggest that this novel configuration did not allow

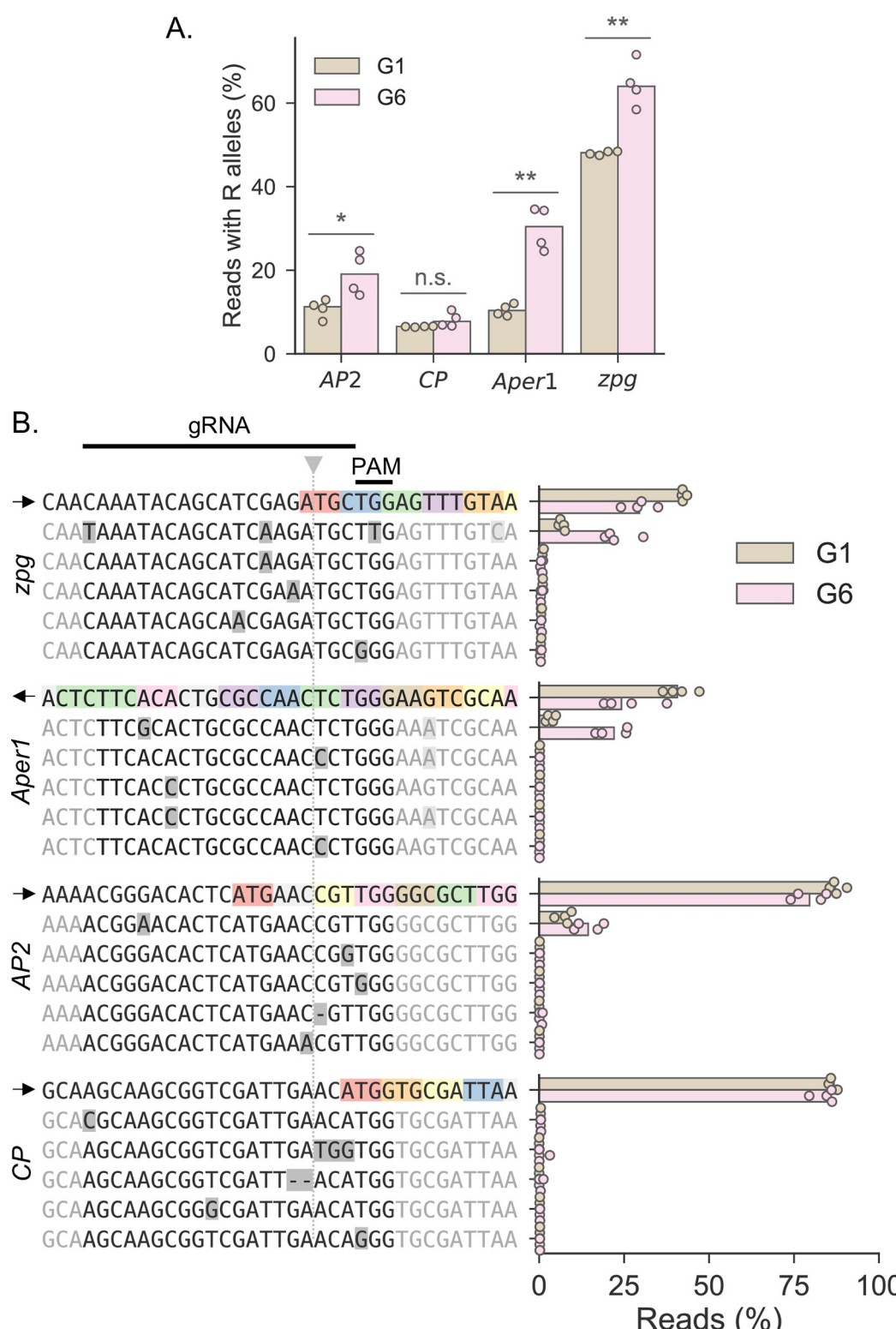

**Fig 5. Analysis of target site variants in the caged populations at all 4 loci. A**: Percentage of reads with a predicted cleavage resistant allele of any type at each locus, at generations 1 (G1, brown) and 6 (G6, pink). Each data point represents a cage population. Paired t-tests were used to compare G1 and G6 at each locus (*AP2*: t = -4.003, p = 0.028; *CP*: t = -1.921, p = 0.151; *Aper1*: t = -7.603, p = 0.005; *zpg*: t = -6.540, p = 0.007) **B**: Shown are the reference allele (top) and the 5 most common target site resistance variants observed at each locus (differences highlighted in grey) and the percentage of reads

mapping to each. Sequences on the left are shown relative to the gRNA (above; cut site shown with grey triangle and dotted line). Coding nucleotides are indicated in colour and the direction of translation is indicated (arrow). The frequency of each of these alleles in G1 and G6 samples is then shown on the right (each data point representing a cage population). The $zpg^R$ allele is shown directly below the reference allele for the $zpg$ data.

for sufficient expression of $zpg$ from the $zpg^D$ allele in females. Alternatively, other transcriptional or translational mechanisms may have led to reduced $zpg$ gene product (e.g. regulatory interference from the Cas9 coding sequence). Read-through translation and the resulting nonsense peptide arising from antisense 3xP3::mScarlet may have also contributed the observed fitness effects. Another method for avoiding gene-regulatory problems associated with marker genes in IGDs is to remove them after transformation using Cre-loxP mediated excision [43] and such an approach could allow the development of neutral drives in the future.

Interestingly, the tendency of $zpg^D$ to cause dominant sterilization in females was counteracted by $zpg$ target site resistance which occurs naturally in lab and wild populations of *A. gambiae*. Thus, in addition to males where both autonomous and non-autonomous gene drive occurred efficiently, the presence of the $zpg^R$ allele in conjunction with $zpg^D$ allowed for the mobilization of unlinked non-autonomous drive elements in females. We hypothesized that due to its sex- and background- specific costs, $zpg^D$ would thus provide a self-removing source of Cas9. Just as gene drives with minimal genetic modifications reduce the changes introduced into individual genomes, approaches reducing the modifications introduced to populations (to only those components necessary for global health, agriculture, or ecology, for example) have obvious appeal. Advances have been made in our ability to recall problematic gene drives by replacing or upgrading them with different transgenes [57,58]. Alternatively, transgenes could be replaced with natural alleles. It has been suggested that gene drive testing pathways for population modification could feature a non-driving source of Cas9 that, having spread non-autonomous payload genes through a population, is slowly lost by selection [26,27]. Although the assumption that simple Cas9-expressing transgenes will impose fitness costs seems plausible, a recent study in *Aedes Aegypti* has shown that, whilst one Cas9-expressing transgene did impose a small fitness cost, another did not have any measurable effect [23]. To build a system where Cas9 is removed unambiguously, an effective, self-eliminating source of Cas9 is desirable. Furthermore, the dynamics of such a transgene must be both modelled, and more importantly, tested in population trials. Recent advances have been made in *Drosophila* where trials showed that non-driving Cas9 transgenes are able to impose a fitness cost in the presence of their non-autonomous payloads by targeting the payloads to essential genes [24]. However, a system where the cost of the Cas9 is unlinked to payloads it is mobilizing has not been described and is not available in medically relevant target species such as *A. gambiae*.

Because the active homing of $zpg^D$, or its leaky somatic expression, results in female sterility, resistant alleles at the $zpg$ target site should be rapidly selected for. Such a resistant allele (referred to as $zpg^R$) already exists in lab and wild populations. When released into a population with $zpg^R$ alleles, $zpg^D$ was expected to undergo strong counterselection, leaving non-transgenic resistant alleles behind. We were able to recapitulate this dynamic in a set of population cage experiments, whilst for the first time, studying the behaviour of multiple interdependent drive elements. These cage trials allowed us to show experimentally that $zpg^D$ can spread non-autonomous drives whilst itself being lost from a population. By sequencing non-transgenic alleles, we also analysed the rise of resistant alleles at each of the 4 loci. We found that whilst the ScoG-AP2 transgene showed the most dramatic spread, it also had the greatest increase in target site resistance over time. On the other hand, Sco-CP showed a ~4-fold increase in frequency, yet no significant increase in resistant alleles over the course of the

experiment. The reasons for this are not obvious. Carboxypeptidase proteins are thought to be highly important for mosquito blood meal digestion [59,60] and our results could indicate functional constraints on the target sequence near the start codon at this locus. Lower levels of pre-existing variation at the target site (although this too could reflect strong functional constraint), or a lower cost of transgene integration could also play a role.

Multigenerational trials using caged populations can tell us a great deal about a gene drive system, but there are fundamental aspects of field populations that cannot be recapitulated in a lab setting. Whilst our simple discrete-generation model goes some way to predicting the behaviour of zpg$^D$ and interdependent transgenes, important aspects of population genetics could not be explored or have not been explicitly investigated. For example, several features accompanying a possible zpg$^D$ release could lead to a reduction in the effective population size. These include the strong selection for resistance alleles, the imbalances in reproductive output between transgenic males and females, or between transgenic and wild-type individuals, and the reduction in census population size due to zpg$^D$'s suppressive effect. This last point is particularly important as it also affects migration (which, in-turn, can influence the effective population size). For example, the smaller a target population is relative to a neighbouring population, the more immigrants it will receive per emigrant, with immigrants also making up a larger proportion of the population [61]. Such an effect is evident in stochastic, spatially-explicit modelling of so-called "reduce and replace" systems [62], where population size is reduced by a female-specific lethal transgene whilst an anti-pathogen payload is transmitted through surviving transgenic males [62,63]. It has been suggested that, with a greater level of population suppression, the effect of wild-type migrants on transgene dynamics is amplified, impacting the long-term frequency of the payload gene. Parallels between this approach and zpg$^D$-based population modification would suggest that we may find similar effects in the early stages of a release where zpg$^D$ is at a high frequency and the population is being suppressed. On the other hand, this asymmetry could be beneficial for migration in the other direction i.e. reduced escape of transgenes and invasion of non-target populations (although note that others have suggested that in many cases, escape is still possible with relatively low levels of migration [64]). Lastly, any effects of, or from, migration will depend strongly on the frequency of resistant alleles in neighbouring populations [65]. Future work will need to determine the importance of these factors in influencing the dynamics of released gene drives.

In summary, by combining a self-eliminating source of Cas9 with minimal, marker-free payload genes, we were able to show the efficacy of a gene drive system that moves towards more minimal modifications of populations following a gene drive intervention. We also show that the 5' of CP is a promising target for non-autonomous integral gene drives due to its favourable profile for the accumulation of resistant alleles. Our population trials also demonstrate the feasibility of multiplexing non-autonomous payloads. The benefits of this are potentially manifold. Firstly, although every payload transgene used here carried the same anti-parasite effector molecule (Scorpine [51]), different effectors could be used in the future in an approach akin to combination therapy. Synergistic interactions between effectors within a mosquito could then lead to an increased potency compared to one effector alone. Furthermore, this combination therapy would reduce the likelihood of resistance evolving in parasite populations. As well as expanding the arsenal of effector molecules, multiplexing could utilise different effector expression profiles, allowing users to target different stages of the parasite lifecycle in the same release [47]. An additional benefit to multiplexing effectors could be the reduced risk posed by unidentified mechanisms of escape or resistance to one transgene or its effector in a wild population. It is possible however, that when multiplexing non-autonomous drives, the level of Cas9 may become limiting [66] if multiple transgenes are present in the same individual.

The cage experiments we performed also mirror another important scenario: suppression drives that have engendered resistance and are failing. Unlike suppression drives, gene drives intended for population modification are typically not designed to carry any fitness cost [53,54]. Because they provide little relative advantage over the drive allele, selection for functional resistant alleles in such drives should be weak. This is evidenced from recent cage trial data in mosquito population modification strategies where resistance alleles had little impact on the performance of the drive [67]. On the other hand, functional resistant mutations pose a large threat to suppression drives [2,19] and have been found to undergo strong selection in cage trials where the drive is costly [41,68]. Whilst various improvements have been made to suppression drives to reduce the likelihood of resistance arising, including the use of highly constrained gRNA target sites [21] and optimisation of the level and spatiotemporal selectivity of Cas9 expression [22,69], the generation and accumulation of resistant alleles still poses a risk to the long-term success of suppression programmes [70,71]. The fitness cost imposed by $zpg^D$ is comparable to (due to its dominance) that of a suppressive gene drive. Accordingly, in the egg output of our cage trials we found that at higher allele frequencies, $zpg^D$ had a suppressive effect on the populations (**S2 Fig**). Furthermore, our cage trials were seeded with standing variation that included functional resistant alleles ($zpg^R$) like those that could be generated by a suppression drive. The dynamics of $zpg^D$ in our cage trials therefore also mimics a failing suppression drive. Our results suggest that if such dynamics were found after suppression programmes have begun, non-autonomous drives could be released to piggyback on residual Cas9 in the population. As malaria resurgence in a human population with reduced levels of immunity is a potential negative outcome of an unsuccessful suppression trial the ability to switch to a population modification strategy, making use of the existing Cas9 alleles in the population, is an attractive fallback strategy.

## Methods

### Target choice and construct design

In an initial attempt, we had designed a simple drive module at the *zpg* locus, consisting of a Cas9-2A followed by a GFP-2A. When integrated immediately before the start of a germline-expressed CDS, this construct should lead to the production of Cas9, GFP, and the endogenous target gene's protein product in the germline. A gRNA targeting the wild-type locus also needed to be included to allow the module to home itself. A synthetic intron was therefore inserted in the Cas9 sequence, containing an *Anopheles gambiae* U6-driven gRNA. This meant that the gRNA could be transmitted along with the rest of the knock-in during HDR, without disrupting translation of the Cas9-2A-GFP-2A knock-in. The sequence surrounding *zpg*, along with its gene annotation [72], was obtained from Ensembl directly through Benchling (AgamP4 genome assembly; accessed 2019). Again, using Benchling, gRNAs targeting the region immediately surrounding the start codon were chosen to determine the knock-in location. Only gRNAs whose target sequence would be disrupted by a clean knock-in directly at the start codon were chosen. However, despite repeated rounds of injections none of the mosaic G0 mosquitoes gave rise to evident transgenic lines. Attributing this to low fluorescent signal from the 2A-GFP (mosquito *zpg* has not been characterised using GFP; furthermore, 2A skipper peptides may also reduce the expression of subsequent CDS due to inefficient cleavage [55,56]), we used the GFP as a vessel for a new bright, dominant, fluorescent marker, codon-optimised for *A. gambiae* (3xP3::AgScarlet). This new marker cassette contained intron donor and acceptor sites so it could be spliced from the GFP coding sequence, but was also re-coded to allow translational readthrough (and generation of a nonsense amino acid chain on the end of the GFP in the event that splicing did not occur–see below). Using these new constructs, we obtained the two transgenic mosquito lines that were investigated here.

## Generation of constructs

A consensus sequence of canonical U2-dependent introns [73] was used to guide the design of a synthetic gRNA-expressing intron within Cas9. We used a human codon-optimised version of Cas9 [45], and the U6::gRNA cassette (containing a spacer-cloning site) was taken from p165 [4]. Splicing donor/acceptor signals were introduced by PCR. Using Gibson assembly, the intron was integrated at an appropriate site within the Cas9 CDS, the whole gene flanked with BbsI sites and integrated within a backbone from pBac[3xP3-DsRed]β2-eGFP(F2A) I-PpoI [74] producing the plasmid, pCas9-INT. After digestion with BbsI, this Cas9{U6:: gRNA} fragment was fused at its 3' end with a T2A sequence [56] (codon-optimised for *A. gambiae* and generated by annealing oligos), followed by a GFP-F2A sequence amplified from pBac[3xP3-DsRed]β2-eGFP(F2A)I-PpoI. This whole fragment was again flanked at each end by BbsI sites and assembled into a backbone amplified from pAmpR_SDM (an unpublished plasmid where a BsaI site has been removed from the AmpR cassette with a silent mutation) to give the plasmid, pUDCas9. A 3xP3::mScarlet fluorescent marker [75] codon-optimised for *A. gambiae*, flanked by splicing donor/acceptor signals, and recoded to allow transcriptional and translational readthrough in the antisense direction was synthesised by ATUM. This fragment was then integrated in the antisense direction into the GFP of pUDCas9 after BsrGI digestion, yielding pUDCas9_v2X. The two gRNAs targeting *zpg* were synthesised as oligos (see primer table) and cloned in by BsaI-based Golden Gate Assembly into the U6::gRNA spacer cloning site of pUDCas9_v2X, yielding pUDCas9_v2X_ZPG1 and pUDCas9_v2X_ZPG2. These plasmids were then digested with BbsI, and the Cas9{U6::gRNA}-2A-GFP{3xP3::AgScarlet}-2A fragments assembled within ~2kb homology arms to the 3' and 5' (amplified from *A. gambiae* genomic DNA), and a universal vector backbone fragment amplified from pBac[3xP3-DsRed] β2-eGFP(F2A)I-PpoI. These plasmids were called pZPGCas9Dr_v2_ZPG1, and pZPGCas9Dr_v2_ZPG2 (GenBank files for these plasmids are provided as **S1 DNA file**). All primers used for generating constructs are listed in **S1 Table**.

## Microinjection of embryos, identification and confirmation of transgenic mosquitoes

Freshly laid G3 embryos were used for microinjection, as described elsewhere [76,77]. Plasmids were purified with endotoxin-free reagents (EndoFree Plasmid Maxi Kit, Qiagen) and all template plasmids were sequenced prior to injection. Injections were performed using 300ng/μl of helper plasmid and 300ng/μl of template plasmid. Surviving $G_0$ larvae showing mosaic RFP expression at the posterior end from the 3xP3 marker in the helper or template plasmids were crossed to wild-type mosquitoes and $G_1$ progeny were screened for positive transformants based on the relevant fluorescent marker (germline-expressed GFP for v1 constructs, 3xP3::AgScarlet for v2). Knock-in of constructs was confirmed by PCR using primers annealing in the genome, beyond the edge of homology arms (see primer table). pZPGCas9Dr_v2_ZPG1, and pZPGCas9Dr_v2_ZPG2 gave rise to $zpg^D_{gRNA1}$ and $zpg^D_{gRNA2}$, respectively.

## Estimation of allele frequencies

The allele frequencies of the SNPs within the $zpg^S$ and $zpg^R$ alleles were estimated using population genetic information from the following sources: Illumina whole genome sequencing (WGS) of 24 G3 individuals [41]; Sanger sequencing at the *zpg* locus (see primer table) of 24 Ngousso individuals; WGS data from 60 Angolan and 44 Kenyan mosquitoes (*Anopheles gambiae* 1000 genomes project [50], phase 1 AR3 data release, accessed July 2020).

## Containment of gene drive mosquitoes

Mosquitoes were housed at Imperial College London in a Level 2-compliant insectary under previously established Arthropod Containment Guidelines [78] and all work was performed under institutionally approved biosafety and GM protocols. As well as the ecological containment provided by the temperate Northern climate of the United Kingdom, six doors separated any cubicles housing gene drive mosquitoes to the external environment and two levels of security access were required to gain entry to these cubicles. These containment steps are compliant with safeguarding guidelines set out by Akbari and colleagues [28].

## Mosquito husbandry

All mosquitoes were maintained at ~26˚C and ~70% humidity with a 12h:12h light:dark cycle and provided with 10% Glucose *ad libitum*. Females were blood-fed with human blood. All sexing was performed at pupal stage. To maintain the $zpg^D$ transgenic line, hemizygous males were backcrossed to G3 females every generation, *ad infinitum*. Isogenic $zpg^R$ and $zpg^S$ lines were obtained by performing a large number of single crosses between G3 males and females. Parents were then genotyped by Sanger sequencing of the gRNA target region (see primer table). An equal number of crosses whose parents were enriched for either $zpg^S$ or $zpg^R$ were then chosen, and single crosses performed again between F1 male and female offspring from different clutches. F2 offspring with two homozygous $zpg^S$ or $zpg^R$ parents were pooled, giving rise to the isogenic $zpg^S$ and $zpg^R$ wild-type lines, respectively. These lines were used only for seeding cage trials.

## Assessing zpg^D in different genetic backgrounds

Due to the presence of the $zpg^R$ allele in the G3 lab colony, our initial maintenance of the $zpg^D$ line (see above) meant that hemizygotes could carry either a $zpg^R$ or $zpg^S$ allele at their wild-type chromosome. $zpg^D$ transmission was therefore assessed using single crosses of hemizygotes to G3 individuals, followed by genotyping of the $zpg$ locus at the wild-type chromosome. For males, crosses were performed in cups by giving a single male three G3 females. These females were then blood-fed and allowed to lay in a fresh cup containing ~25mL water and lined with filter paper. Female hemizygotes were allowed to mate with G3 males and blood-fed *en-masse*, before being distributed to single cups and allowed to lay, individually. The larvae (minimum of 20, mean ± std of 50.9 ± 17.9) of each cross were then screened for the presence of the $zpg^D$ allele based on red fluorescence using an Olympus MVX10 stereo microscope. Parents from each single cross were transferred to PCR tubes and the $zpg$ allele at their wild-type chromosome was amplified using Phire Tissue Direct PCR Mastermix (Thermo Scientific; see primer table). Amplicons were then sent for Sanger sequencing. Sequencing files were analysed in Benchling.

To assay transmission of a non-autonomous gRNA-only construct in the presence of $zpg^D$, we generated double hemizygote $zpg^D$, Sco^GFP-CP [43] individuals by crossing hemizygote $zpg^D$ males to homozygote Sco^GFP-CP females and selecting any doubly fluorescent mScarlet, GFP offspring. Double hemizygote males or females were then crossed to G3 mosquitoes of the opposite sex as above, and females were allowed to lay, individually. Larvae from each cross (minimum of 21, mean ± std of 45.9 ± 8.1) were then screened for red and green fluorescence using an Olympus MVX10 stereo microscope. The data for plotting has been provided in file S1 Data .

The *in-vitro* cleavage assay was performed using PCR products at the $zpg$ locus containing either the $zpg^S$ or $zpg^R$ allele, obtained during homing assays (above, see primer table). gRNAs were synthesised *in-vitro* using the TranscriptAid T7 High Yield Transcription Kit (Thermo

Scientific) and a template. The template was generated by PCR using the following oligos: gRNA-InViTr_ZPG1 or gRNA-InViTr_ZPG2 (0.02µM); gRNA-InViTr_scaffold (0.02µM); gRNA-InViTr_fwd (2µM); gRNA-InViTr_rev (2µM). Templates were then purified using the QIAquick PCR Purification Kit (Qiagen). *In-vitro* transcription reactions were incubated for 8 hours at 37˚C, treated with DNase I (NEB), and purified using the RNeasy MinElute Cleanup Kit (Qiagen). Cleavage reactions were performed using 3nM of *zpg* PCR fragment, 45nM of sgRNA and 45nM of *S. pyogenes* Cas9 nuclease (NEB). After heat-inactivation at 65˚C, the entire volume of cleavage reaction was loaded and resolved using gel electrophoresis.

## Phenotypic characterisation of zpg genotypes

To compare the fertility of homozygous $zpg^D$ males and females to hemizygotes, we first obtained hemizygous and homozygous individuals by crossing hemizygous males and females *en-masse*. Note that because $zpg^D$/$zpg^S$ females are not fertile (see below), all hemizygous off-spring carried a $zpg^R$ allele in their genetic background. The F1 of this cross were then separated into single cups and each given one G3 mate of the opposite sex. After 5 days, females were blood-fed and transferred to fresh cups to lay. After three days, eggs were counted for each female, and females were collected. Any females that did not lay viable eggs were dissected and their spermathecae examined. Unmated females were excluded from further phenotypic analysis. The number of larvae hatching from the eggs was counted over a few days. 46 transgenic male and 46 transgenic female crosses were analysed in total, of which 32 male and 10 female crosses either were unsuccessful or could not be genotyped.

To genotype hemizygotes vs homozygotes, genomic DNA was extracted from individual transgenic mosquitoes using DNeasy Blood & Tissue kits (Qiagen). PCR was then performed on each sample using two primer sets: one spanning the 5' junction of the knock-in, with one internal and one external primer; and one spanning either side of the integration site (see primer table). The first primer set produces a band in the presence of a transgene, the second produces a band in the absence of one. Theoretically, this second primer set also produces a very large band in the presence of a transgene, however the predicted amplicon is too large (>7.5kb) given the polymerase and extension times used for genotyping, and this band was not obtained. Using a specialist polymerase for PCR with long amplicons (LongAmp Taq, NEB), a handful of DNA samples were randomly chosen and confirmed using a third primer set spanning the whole transgene with primers annealing beyond the edge of the homology arms used for the initial knock-in (Fig 3; see primer table).

To quantify the size of reproductive organs from mosquitoes of different genotypes, hemizygous $zpg^D$ individuals were intercrossed and virgin male and female F1 offspring, along with virgin male and female G3 mosquitoes, were dissected straight into PBS. Testes and ovaries were then imaged on an inverted EVOS FL Imaging System without coverslips and carcasses were genotyped by Sanger sequencing (see above). The Length of testes and ovaries were estimated in ImageJ by tracing a segmented line through the centre of each organ. Testes area was estimated in ImageJ by approximating each testis with an ellipse, then taking the squareroot. A minimum of 9 mosquitoes were dissected of each genotype (mean ± std of 15.1 ± 3.8). The data for plotting has been provided in file **S1 Data**.

## Cage trial assay

We carried out trials in four independent cage populations. The double hemizygotes used to seed the cages were obtained by crossing hemizygous $zpg^D{}_{gRNA1}$ males with homozygous females of each effector line (ScoG-AP2, Sco-CP or *Aper1-Sco*). Individual females from each cross were then allowed to lay, before having their wild-type *zpg* allele genotyped by sanger

sequencing. The majority of ScoG-AP2 and Sco-CP homozygotes were $zpg^S/zpg^S$, and only the progeny of $zpg^S/zpg^S$ females were kept. $Aper1$-$Sco$ homozygotes were uniformly $zpg^R/zpg^R$. Most likely this is due to the $Aper1$-$Sco$ line being founded in a $zpg^R$ background and the two loci are in proximity (2L, 38.6Mb and 2L,28.6Mb, respectively). The progeny of each cross was maintained separately until pupae, then screened for the $zpg^D$ mScarlet marker and sexed. The resulting double hemizygote males were then used to initiate cage trials in 24.5cm$^3$ cages. Each cage was initiated using: 60 $zpg^D_{gRNA1}/zpg^S$, ScoG-AP2 double heterozygote males; 60 $zpg^D_{gRNA1}/zpg^S$, Sco-CP double heterozygote males; 60 $zpg^D_{gRNA1}/zpg^R$, $Aper1$-$Sco$ double heterozygote males; 120 wild-type $zpg^S/zpg^S$ males and 300 wild-type $zpg^S/zpg^S$ females. Mosquitoes were left to mate for five days before being fed on human blood. Two days later, mosquitoes were given egg bowls consisting of conical filter paper (125mm diam., Whatman) submerged in 300ml water for oviposition. Eggs were spread evenly across the filter paper with water from a Pasteur pipette and photographed. Images were then used to estimate egg numbers using the MosceggApp egg-counting software [79] with egg-detection sensitivity set to minimum and non-egg detection set to approximately 66%. 600 larvae emerging from these eggs were then randomly chosen to seed the next generation and maintained at 150 larvae per pan until pupation. At each generation, once adults had been allowed to lay, 47 were randomly collected from each cage and genotyped, along with one wild-type per-cage as a negative control, using multiplex-PCR (below). A further 100 were collected for pooled DNA extraction and amplicon sequencing (below).

We required a molecular method for genotyping. A multiplex PCR approach was optimised using allele-specific primers that amplify fragments for the transgenic, but not wild-type alleles, at each locus. Whole mosquitoes were lysed in separate wells of 96-well plates and immediately used for PCR following the Phire Tissue Direct PCR Master Mix (Thermo) Dilution & Storage protocol. As well as primer sets at the four transgenic loci, we also included a primer pair at the S7 ribosomal subunit gene as an internal positive control for each PCR. The same reverse primer within Scorpine was used in all effector transgene primer sets. Primers were designed to produce a discernible banding pattern that would allow discrimination between each locus, with the S7 control product giving the largest band. Any individuals that didn't produce this larger band were ignored. Various primers and combinations were tested, and the annealing temperatures of the best primer set was optimised. For each transgenic locus, the specificity of its final primer pair was confirmed by performing PCR with transgenic DNA samples from the other three loci. The combined primers for all transgenes also produced no bands when using wild-type DNA as a template. Although a primer pair was included in the set for benchmarking, $zpg^D$ was genotyped using its 3xP3-driven fluorescent reporter. See **S1 Data** file for individual cage data.

## Amplicon sequencing

Primers were designed to amplify 300-500bp regions surrounding the genomic gRNA target site at each locus. Primers were given a 5mer barcode at the 5' end specific to the generation (forward primer) and replicate (reverse primer), allowing amplicons from different DNA samples to be multiplexed (see primer table). Genomic DNA was extracted from pools of 100 mosquitoes per replicate at two generations (G1 and G6). For PCR, KAPA HiFi HotStart polymerase (Roche) was used with 25 cycles. Amplicons were purified using Qiaquick PCR purification columns (Qiagen) and quantified using Qubit (ThermoFisher). Thirty-two samples were multiplexed in total (2 generations, 4 replicates, 4 loci). 5ng of DNA from each sample was pooled (160ng total) and sequenced on one MiSeq lane with 2x300bp reads (Eurofins Genomics).

Je [80] was used to initially de-multiplex reads using the 5' barcode, detailing the generation of the sample, and the 3' barcode, detailing the replicate. CRIPSPResso2pooled [81] was then used to merge read pairs and map them to the four different amplicons (default parameters were used, apart from—max_paired_end_reads_overlap which was set to 252 – 10bp more than the predicted overlap at the smallest amplicon). Any read pairs unassigned a sample by Je, were then demultiplexed using fastx-toolkit [82] from the single-end flash-merged reads, where sequencing errors present towards the ends of separate reads have been reduced. These were merged with their corresponding fastq file using bash. The thirty-two resulting fastq files were then re-mapped and analysed using CRISPResso2bulk (with the same parameters used for CRISPResso2pooled; see above). Mapping statistics are provided in **S2 Table**. Reads were deposited in the European Nucleotide Archive (ENA) under accession number PRJEB50687.

Using the frequency tables of alleles around the sgRNA produced by CRISPResso2, a custom Python script was used to select only those alleles with mutations in the region bound by the gRNA spacer or the PAM (i.e. those that would potentially cause resistance to gene drive). For the first analysis in Fig 5A, the frequency of all these alleles were summed for each sample, then this sum calculated as a percentage of the total number of reads for that sample. For the second analysis, the five most common alleles (across all samples and replicates) were selected. Whether a given allele was absent in any G1 samples was also calculated, and that allele given a frequency of 0 for that sample.

## Supporting information

**S1 Fig. Discrete generation population model of $zpg^D$ interacting with 3 non-autonomous payload drives.** A: Stochastic simulation of populations of 600 adult mosquitoes with a starting population of 300 wild-type females, 120 wild-type males (all $zpg^S$) and 60 $zpg^D/zpg^S$, Sco-CP/+ males; 60 $zpg^D/zpg^S$, ScoG-AP2/+ males and 60 $zpg^D/zpg^S$, Aper1-Sco/+ males. Shown are the average transgene frequencies at each locus calculated from 47 randomly sampled individuals out of 25 simulated populations. Because of this sampling strategy, which mimics our experimental one, starting transgene frequencies of zpgD (30%) and each of the payloads (10%) in individual populations may differ from the mean across populations (black line). B: The number of individuals of each indicated genotype at the 4 loci of the simulations in A of all 600 individuals in each population. Drive here refers to both autonomous ($zpg^D$) and non-autonomous (Sco-CP, Aper1-Sco, ScoG-AP2) elements. C: Simulations as in A but assuming the presence of pre-existing R1 target site resistance at the $zpg$ locus. These were introduced as 60 $zpg^D/zpg^R$, Aper1-Sco/+ males resulting in 5% starting allele frequency of $zpg^R$. Shown are the average transgene frequencies at each locus calculated from 47 randomly sampled individuals out of 25 simulated populations. Because of this sampling strategy, which mimics our experimental one, starting transgene frequencies of zpgD (30%) and each of the payloads (10%) in individual populations may differ from the mean across populations (black line). D: The number of individuals of each indicated genotype at the 4 loci of the simulations in C of all 600 individuals in each population. Drive here refers to both autonomous (zpgD) and non-autonomous (Sco-CP, Aper1-Sco, ScoG-AP2) elements.
(PDF)

**S2 Fig. Changes in population reproductive capacity during spread of transgenes. A**: Total egg output of each caged population, at each generation shown in light grey; mean of all cages at each generation shown in black; predicted trajectory given no change in reproductive output (mean of all cages at G0) shown by dashed red line. **B**: The integral of each trajectory (grey circles) and their mean (grey bar) compared to the integral of the no-change trajectory (red dashed line). A one-sample t-test was used to compare the integrals of each real timeseries to

that of the predicted no-change timeseries (t = -13.2, p<0.001).
(PDF)

**S1 Table. List of primers used.**
(XLSX)

**S2 Table. Mapping statistics for the amplicon sequencing experiment.**
(XLSX)

**S1 Data. Raw data for the generation of Figs 2,3 & 4.**
(XLSX)

**S1 Methods. Description of the gene drive model.**
(DOCX)

**S1 DNA file. Annotated DNA files for transformation constructs pZPGCas9Dr_v2_ZPG1 and pZPGCas9Dr_v2_ZPG2.**
(ZIP)

## Acknowledgments

We would like to thank Andrew Hammond and Roberto Galizi for providing constructs and useful discussion; Tony Nolan for providing NGS data of the G3 colony; Paolo Capriotti for occasional assistance with mosquito rearing; Alexander Nash and Andrea Beaghton for initial discussions on modelling; Astrid Guth for help optimising the multiplex PCR for genotyping individuals from cage trials; and Maria Giorgalli for phenotypic characterisation of the $zpg^D$ transgene that was not used in the final manuscript.

## Author Contributions

**Conceptualization:** David A. Ellis, George K. Christophides, Nikolai Windbichler.

**Data curation:** David A. Ellis.

**Formal analysis:** David A. Ellis, Nikolai Windbichler.

**Funding acquisition:** George K. Christophides, Nikolai Windbichler.

**Investigation:** David A. Ellis, George Avraam, Astrid Hoermann, Claudia A. S. Wyer, Yi Xin Ong.

**Methodology:** David A. Ellis, George Avraam, Nikolai Windbichler.

**Project administration:** George K. Christophides, Nikolai Windbichler.

**Resources:** Astrid Hoermann, George K. Christophides, Nikolai Windbichler.

**Software:** Nikolai Windbichler.

**Supervision:** Nikolai Windbichler.

**Visualization:** David A. Ellis, Nikolai Windbichler.

**Writing – original draft:** David A. Ellis, Nikolai Windbichler.

**Writing – review & editing:** David A. Ellis, George K. Christophides, Nikolai Windbichler.

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
