## [Decision Letter · Decision Letter 0]

30 Mar 2022

Dear Dr Windbichler,

Thank you very much for submitting your Research Article entitled 'Testing non-autonomous antimalarial gene drive effectors using self-eliminating drivers in the African mosquito vector Anopheles gambiae' to PLOS Genetics.

The manuscript was fully evaluated at the editorial level and by independent peer reviewers. The reviewers appreciated the attention to an important topic but identified some concerns that we ask you address in a revised manuscript

We therefore ask you to modify the manuscript according to the review recommendations. Your revisions should address the specific points made by each reviewer.

[LINK]

Yours sincerely,

Jackson Champer

Associate Editor

PLOS Genetics

Gregory P. Copenhaver

Editor-in-Chief

PLOS Genetics

Dear Dr. Windbichler and Colleagues,

This manuscript has been examined by three reviewers and myself, and we all agree that it will be most likely suitable for publication in PLoS Genetics, though some revisions will be needed.

Please attempt to address all reviewer comments. In particular, pay attention to data accessibility (both for raw data and plasmid maps) and the extra (?) supplemental table.

Recommended Additional Revisions:

The authors assume that zpg is required in germline cells, so drive/wild-type individuals become sterile by germline drive conversion. More likely than not, this is correct, but it dismisses another possibility that could explain all or some of the sterility. Based on previous studies of zpg-Cas9 in suppression drives, it is known that zpg leads to some “leaky somatic” expression in somatic cells, which moderately reduces the fertility of females (by cutting the wild-type allele). It might be that this is an important factor in reducing zpg expression in important cells, so the possibility should at least be mentioned.

For the discussion about Figure 5, I think it might be helpful to take a step back and introduce for a more general audience what makes a functional resistance allele and why seeing these at high frequency is a pretty good reason to assume that they are r1 (compared to r2). It also seems a little odd to me that more possible r2s were not detected in G1 (compared to G6). Maybe there was a mechanism to quickly eliminate them.

The modeling section in the methods should be considerably expanded. Including the code is good, but the methods should still fully describe the model. If needed, it can be a supplemental methods section.

Minor Comments and Suggestions (revisions at the discretion of the authors):

I liked reviewer 2’s comment on population replacement versus population modification. I agree with the terminology myself as well.

Please take care to include line numbers or at least page numbers when submitting to journals. Some add it automatically, but this copy was not marked with line or even page numbers.

Author affiliation #3 does not seem to be used.

Split drives are reference in the introduction [23,24,26,32,33], but this actually underestimates how widely they are used for biosafety or other reasons. Here are some more (this might even be missing some):

https://elifesciences.org/articles/41439 (first split drives in insects)

https://www.liebertpub.com/doi/10.1089/crispr.2021.0129

https://www.pnas.org/doi/10.1073/pnas.2004373117

https://www.sciencedirect.com/science/article/pii/S2211124720308226

https://journals.plos.org/plosbiology/article?id=10.1371/journal.pbio.3001478 (I think the other two mouse papers also used split drives)

https://www.nature.com/articles/s41467-020-14960-3

The long-term behavior of the cage populations would be interesting, such as where the final equilibrium values fall past generation ten. If the authors have kept the cages going, it would be nice to include this data in the manuscript (even just a few more generations), but this is certainly not necessary.

Sincerely,

Jackson Champer

Reviewer's Responses to Questions

**Comments to the Authors:**

Reviewer #1: In this paper, the authors present the testing of the direct integration of a Cas9 drive cassette at the zpg locus in Anopheles gambiae, with a novel readthrough strategy to minimise modifications. They go on to investigate the impacts of resistance alleles in a target population, analysing homing rates, fitness consequences and drive dynamics to test the efficacy of a system as a proposed non-autonomous drive.

The paper is technically very interesting, well written and clear throughout, and potentially of significant importance in the continued development and refining of gene drive systems for pest control.

I summarise a list of primarily minor comments that I hope will improve the manuscript for publication.

Main comments:

I have some concerns about data management that I hope are easily fixable by the authors. I cannot find the raw data to accompany fertility assays and cage trials; I also cannot see a link to a data repository. Supplementary Table 1 appears to be infection data, the analysis of which I cannot see mentioned anywhere in the manuscript.

The authors summarise one of the uses of this drive as "Self-eliminating drivers could thus enable local population modification without leaving

behind any driver element." However - in this scenario the zpg drive has a suppressive effect on the population. As effector frequency rises within a reducing effective population size, the effect of migration into the release area will change the drive's dynamics. I would like to see the authors discuss the potential implications of migration even if it is not formally incorporated into any models.

Minor comments

Transmission of zpgD and a payload in susceptible and resistant genetic backgrounds:

The authors introduce the Sco (Scorpine)-CP(Carboxypeptidase) line, this is used in a nice experiment to demonstrate that lack of homing of the zpg cassette is due to resistance at that locus and not lack of Cas9 activity; but while it is referenced in another paper it would be nice if the authors could briefly expand into the full names of these constructs and loci here, especially as they mention Scorpine in full elsewhere.

Fertility and gonad development of zpgD transgenics:

In this section the authors make multiple comparisons between homozygous/heterozygous males and females. The experiments are clear, but some aspects of the analysis need clarity or could be improved. There are many pairwise comparisons here

with t-tests (Student's or Welch's? this is undefined), and it is not clearly explained whether these are first tested across one general linear model before diving into these pairwise comparisons. Some of the distributions of data (in some groups there are no offspring) seem ill-suited to a t-test, but I also don't think incorrect conclusions have been drawn here because of the large effect. Although some sample sizes are given at the beginning, it is important to include the degrees of freedom for each t-test as this will determine the critical threshold for statistical significance.

The comparison between male and female transgenics and offspring production does not seem like a 'fair test' I would be surprised if even with sperm depleted males, they could not fertilise all wild-type female eggs - a more thorough comparison would compare them in sperm competition with wild-type males.

When discussing gonad morphology, the measure for ovary atrophy is not described.

Population cage dynamics of zpgD in combination with payload drives:

The authors describe the drive of a non-autonomous element in the presence of target-site resistance at zpg as seemingly paradoxical. But I'm not sure why they describe this as such, many split-drive designs have previously been discussed, this seems perfectly logical.

When describing the " introduction of resistant alleles

would induce gene drive and increase effector frequency at the 3 payload loci" - could the authors summarise this effect?

Figures

Figure 1B - label that the grey letters indicate SNPs

Figure 1D - the scale and colours are not the easiest to read.

Reviewer #2: In an effort to enrich the toolbox for gene drive in mosquito vectors of human malaria, the authors sought to establish a driving Cas9 transgene integrated non-disruptively within the coding sequence of the essential zpg mosquito gene, to serve as a Cas9 source for spreading some non-autonomously driving antimalarial transgenes in a target mosquito population. Somewhat unexpectedly, Cas9 insertion at zpg had a large fitness cost as it impaired zpg activity, especially in females, resulting in strong selection for pre-existing zpg alleles resistant to Cas9 cleavage, counteracting the drive. The authors turn this defect into an advantage as it provides a system for transiently sustaining Cas9 in a mosquito population and reach some level of spread of a set of 3 non-autonomous, gRNA-encoding transgenes, before inexorable disappearance of Cas9 due to the rise of non-costly resistant zpg alleles.

I agree with the conclusions of this study and with most discussion points. The concept and experimental demonstration of a full Cas9 gene drive that self-eliminates after some period of activity, on its own and on distant loci, is quite novel (the molecular mechanism is different and more simple than for the previously theorized “daisy drive” strategy). Programmed disappearance of Cas9 could be partially reassuring to some critics of the gene drive technology, even if the driven non-autonomous transgenes (technologically less frightening) could persist. I appreciate the model of malaria control by gene drive mosquitoes using a combination therapy-like approach through multiple independent anti-malarial transgenes, all driven by the same Cas9 locus. The experimental procedures employed to obtain the results (genotyping and combining defined proportion of mosquitoes of desired genotypes in genetic crosses and follow-up of the frequency of 4 transgenes by genotyping 47 individuals over 6 generations) represent a considerable amount of work.

This work will be of interest to the gene drive field (a contemporary hot topic) and to evolutionary geneticists. It will also be read by stakeholders in mosquito control, including the risk assessment community, and more generally by people interested in and/or worried by gene drive and biotechnology at large. I would be happy to see this study published, once the following points have been addressed or discussed:

Introduction: in an effort towards clarifying vocabulary in the gene drive field, I suggest to reserve the term population replacement for gene drives based on underdominance (whereby the genomic background of the released strain should entirely overwrite that of field populations); and population modification for gene drives that introgress novel genetic traits into a field population’s genetic background without altering the bulk of it (as would be the case with the approach presented here). It’s up to the authors to adopt this point of view or not, if so using “population modification” throughout the manuscript.

Introduction third §: I believe the authors mean “to the expense of the drive” instead of “in favour of the drive”. A bit further: note that if a non-driving Cas9 insertion is costly, it should still disappear by natural selection.

Last § of intro: for more clarity write “capable of providing the antimalarial transgenes with a source of Cas9”….

Results: construct design is not immediately clear: it may be useful to mention that the antisense 3xP3-Scarlet reporter gene must be devoid of any stop codon to allow readthrough translation (also in fig 1B legend). In fact I wanted to check for the absence of stop codons in the antisense 2A-3xP3-Scarlet-2A cassette but the nucleotide sequence of the final construct is not provided. Given the complexity of the cloning scheme used to obtain it, providing an annotated sequence file of the whole construct (plasmid or integrated in its genomic context) as a Suppl. file is required. What might be the physiological effect of expressing the nonsense peptide arising from antisense Scarlet and 3xP3 in zpg-expressing cells? It could contribute to the fitness cost.

Result § “transmission of zpgD…”: for clarity add: “to determine their zpg R or S status, hemizygous parents were then genotyped…” To guide the reader could also add: “Figure 2A, purple bars / Figure 2A, yellow bars”

Fig. 1C: I’m not convinced of the necessity of the left panels with chromatograms, redundant with Fig 1B.

Fig. 1D: why not indicate the polymorphic nucleotides and their position in each strain, not only G3. The heat map for allele frequency is visually not very clear (consider blue to red rather than white to red, to make the dynamic range of frequencies more contrasted?)

Fig. 1D I think the correct spelling of the strain name is Ngousso, a suburb of Yaoundé in Cameroon. Also in Materials & Methods.

Fig 3A The purpose of showing PCR products 2a and 3 is not obvious here (they’re not on the associated DNA gel), I didn’t notice an explicit reference to them in the text.

B of figure 3B is missing.

Text referring to figure 3B: speaking of insemination rate suggests that spermathecae were dissected and counted to assess the number of females that received sperm, independently of sperm fertility. I see in the Methods that this was indeed done but here the number of males that successfully generated progeny is discussed => replace “insemination” by “male reproductive success / ability” ?

In the parenthesis referring to Figure 3C,D you could precise “comparing gonad size in each genotype compared to the WT” for more clarity

Suppl Fig 1A: could mention in the legend that all theoretical curves don’t start exactly at the initial G0 frequency of 0.15 for zpgD and 0.05 for each antimalarial transgene due to simulated random sampling (I was initially confused)

Where 3B is cited, it’s a bit awkward to write that “none” is “significantly fewer” than (anything).

Non-significance of the number of progeny sired by zpgD/D compared to zpgD/R is certainly due to the very low sample size for D/D.

End of the Results section: not immediately obvious what is meant by “mutations distal to the predicted cleavage site” → relatively distant from… Note that the last shown zpg mutant allele should not be a resistant allele, as the TGG PAM is simply changed to GGG, also a valid PAM. I wonder if those mutants with a very distal mutation (e.g. first one for CP) would truly block Cas9 cleavage. Do the authors think these distal mutations arose from Cas9 activity, or are pre-existing polymorphisms detected or not in the 1000 genome sequencing project? The more distal they are, the more likely they’re to be pre-existing polymorphisms. If so, such alleles should not be called mutations but SNPs. Is this what the authors wanted to note in the legend of figure 5B ("Note that for some loci (e.g. Aper1) many, low-frequency alternative alleles exist that are neither R alleles nor reference alleles") ? This note is not so clear and it’s hard to know whether a given allele is R or not. Actually, could these polymorphisms simply be PCR mutations that become visible in high-throughput amplicon sequencing?

It’s actually quite amazing that for Aper and AP2 the resistant alleles gaining most frequency at G6 are very distal mutations that look like they should not completely block Cas9 cleavage. This may suggest that even single very distal mismatches efficiently preclude cutting in Anopheles.

Discussion: please rephrase “and the zpgD Cas9 module was linked with the endogenous zpg CDS by a 2Aseparated translational unit representing the fluorescent transformation marker” in a more precise manner. In fact the 2A-flanked translational unit is a nonsense peptide, it doesn’t represent the fluorescent marker but arises from translation of the marker’s reverse-complement coding and regulatory sequences.

The authors speculate that this design involving a 2A-flanked nonsense peptide could explain the reduction of ZPG protein expression. This could indeed be the case, but alternative explanations are possible. The presence of the Cas9 coding sequence itself may be sufficient to suppress zpg expression. Either way, this may be a translational decrease, or even alternative intron splicing involving the synthetic intron’s 5’ splice site in Cas9 and a 3’ splice junction from one of the downstream zpg introns.

One thought to share with the authors, not necessarily to discuss in the paper: in risk assessment of the zpgD system (should it ever be used in the field), one aspect would be to assess whether natural malaria-susceptibility alleles could be genetically linked to zpgR. This would be undesirable as it would increase the frequency of malaria susceptibility genes hitch-hiking with zpgR ! This could be assessed in the lab by comparing vector susceptibility to P. falciparum of final, zpgR-rich populations to their initial, zpgS-rich population precursor. I agree that besides this caveat, the idea of replacing the Cas9 gene drive allele with a pre-existing natural resistance allele has great appeal.

First § of methods could be more straightforward. The meaning of “we used GFP as a vessel for a new fluorescence marker” is not very clear: if I understand correctly GFP sequence was simply substituted for a new fluorescence marker (if GFP sequence fragments are left over, they should be annotated in the Suppl. nucleotide sequence file I requested).

Second § of Methods: worth mentioning whether the 2A-Scarlet cassette has been verified/recoded to be devoid of stop codons in frame with the Cas9-zpg coding sequences, referring to the new Suppl. sequence file.

Eighth § of Methods: not sure I agree with “Note that because zpgD/zpgS females are not fertile (see below), all hemizygous offspring carried a zpgR allele in their genetic background.”

→ Can’t hemizygous offspring carry zpgD from a zpgD/zpgR mother and zpgS from a zpgD/zpgS father?

Should homogenize usage of the terms –gote / gous throughout the text when used in the adjective position (hemizygote/hemizygous , homo- etc. but as a noun keep –gote)

A few typos : One “rapidly” too many on page 11 of the full pdf, hemiozygote in Methods, “to” is missing in “compared one effector alone”, it’s for its …effector in a wild population, singular/plural inconsistency in last sentence of acknowledgements.

Reviewer #3: Uploaded as an attachment

**Have all data underlying the figures and results presented in the manuscript been provided?**

Reviewer #1: **No: **I have some concerns about data management that I hope are easily fixable by the authors. I cannot find the raw data to accompany fertility assays and cage trials; I also cannot see a link to a data repository. Supplementary Table 1 appears to be infection data, the analysis of which I cannot see mentioned anywhere in the manuscript.

Reviewer #2: **No: **Nucleotide sequence of the plasmid knocked into the zpg locus

Reviewer #3: **No: **Annotated Construct Sequences should be deposited in GenBank or analogous resource

PLOS authors have the option to publish the peer review history of their article (what does this mean?). If published, this will include your full peer review and any attached files.

Reviewer #1: No

Reviewer #2: **Yes: **Eric Marois

Reviewer #3: **Yes: **Vanessa M. Macias

---

## [Editor Report · Decision Letter 1]

9 May 2022

Dear Dr Windbichler,

We are pleased to inform you that your manuscript entitled "Testing non-autonomous antimalarial gene drive effectors using self-eliminating drivers in the African mosquito vector Anopheles gambiae" has been editorially accepted for publication in PLOS Genetics. Congratulations!

Yours sincerely,

Jackson Champer

Associate Editor

PLOS Genetics

Gregory P. Copenhaver

Editor-in-Chief

PLOS Genetics

Comments from the reviewers (if applicable):

All the changes look fine here. I think that the article is ready for publication. My only comment is that the supplemental method section could probably use some more paragraph breaks.

Sincerely,

Jackson Champer

**Data Deposition**

http://datadryad.org/submit?journalID=pgenetics&manu=PGENETICS-D-22-00190R1

**Press Queries**

---

## [Editor Report · Acceptance letter]

25 May 2022

PGENETICS-D-22-00190R1 

Testing non-autonomous antimalarial gene drive effectors using self-eliminating drivers in the African mosquito vector Anopheles gambiae 

Dear Dr Windbichler, 

We are pleased to inform you that your manuscript entitled "Testing non-autonomous antimalarial gene drive effectors using self-eliminating drivers in the African mosquito vector Anopheles gambiae" has been formally accepted for publication in PLOS Genetics! Your manuscript is now with our production department and you will be notified of the publication date in due course.

With kind regards,

Livia Horvath

PLOS Genetics

On behalf of:
